# Synergic Effect of *Brachyspira hyodysenteriae* and *Lawsonia intracellularis* Coinfection: Anatomopathological and Microbiome Evaluation

**DOI:** 10.3390/ani13162611

**Published:** 2023-08-13

**Authors:** Amanda G. S. Daniel, Carlos E. R. Pereira, Fernanda Dorella, Felipe L. Pereira, Ricardo P. Laub, Mariana R. Andrade, Javier A. Barrera-Zarate, Michelle P. Gabardo, Luísa V. A. Otoni, Nubia R. Macedo, Paula A. Correia, Camila M. Costa, Amanda O. Vasconcellos, Mariane M. Wagatsuma, Thaire P. Marostica, Henrique C. P. Figueiredo, Roberto M. C. Guedes

**Affiliations:** 1Department of Veterinary Clinic and Surgery, Veterinary School, Universidade Federal de Minas Gerais, Belo Horizonte 130161-970, Brazil; amandavet2007-1@hotmail.com (A.G.S.D.); carlos.pereira@ufv.br (C.E.R.P.); fernandadorella@gmail.com (F.D.); ricardolaub@gmail.com (R.P.L.); mariana.andrade@agro.gov.br (M.R.A.); jabarreraz@unal.edu.co (J.A.B.-Z.); michelle.gabardo@ifmg.edu.br (M.P.G.); luisavianna.vet@gmail.com (L.V.A.O.); medvetpaulacorreia@gmail.com (P.A.C.); camila.mcosta@outlook.com (C.M.C.); vasconcellosamand@gmail.com (A.O.V.); marianemw7@gmail.com (M.M.W.); thairepereira@gmail.com (T.P.M.); 2Department of Preventive Veterinary Medicine, Veterinary School, Universidade Federal de Minas Gerais, Belo Horizonte 130161-970, Brazil; felipe@flpsw.com.br (F.L.P.); figueiredoh@yahoo.com (H.C.P.F.); 3College of Veterinary Medicine, Iowa State University, Ames, IA 50011, USA; nubia@iastate.edu

**Keywords:** proliferative enteropathy, swine dysentery, experimental infection, bacterial profile, 16S sequencing

## Abstract

**Simple Summary:**

Clinical and anatomopathological evaluations of animals experimentally inoculated with *Brachyspira hyodysenteriae* and 7 days later with *Lawsonia intracellularis* were performed and compared to single-infected pigs. Co-infected animals were more affected and had more severe lesions when compared to single-infected pigs. The intestinal microbiome of pigs from co-infected animals demonstrated a difference in some genera compared to other groups before and after inoculation and certain genera were evaluated in each group.

**Abstract:**

*Brachyspira hyodysenteriae* and *Lawsonia intracellularis* coinfection has been observed in the diagnostic routine; however, no studies have evaluated their interaction. This study aimed to characterize lesions and possible synergisms in experimentally infected pigs. Four groups of piglets, coinfection (CO), *B. hyodysenteriae* (BRA), *L. intracellularis* (LAW), and negative control (NEG), were used. Clinical signals were evaluated, and fecal samples were collected for qPCR. At 21 days post infection (dpi), all animals were euthanized. Gross lesions, bacterial isolation, histopathology, immunohistochemistry, and fecal microbiome analyses were performed. Diarrhea started at 12 dpi, affecting 11/12 pigs in the CO group and 5/11 pigs in the BRA group. Histopathological lesions were significantly more severe in the CO than the other groups. *B. hyodysenteriae* was isolated from 11/12 pigs in CO and 5/11 BRA groups. Pigs started shedding *L. intracellularis* at 3 dpi, and all inoculated pigs tested positive on day 21. A total of 10/12 CO and 7/11 BRA animals tested positive for *B. hyodysenteriae* by qPCR. A relatively low abundance of microbiota was observed in the CO group. Clinical signs and macroscopic and microscopic lesions were significantly more severe in the CO group compared to the other groups. The presence of *L. intracellularis* in the CO group increased the severity of swine dysentery.

## 1. Introduction

Enteric diseases are commonly related to a single microorganism; however, the microbiota composition plays an essential role in establishing an infection [1,2]. Changes in microbiota are constantly modulated by pH, temperature, and nutritional conditions [1,3]. Swine dysentery and proliferative enteropathy are among the most common enteropathogens in growing and finishing pigs [4].

Swine dysentery (SD) is caused by three Brachyspira species, the most common of which is *Brachyspira hyodysenteriae*, a fastidious Gram-negative anaerobic spiral-shaped bacterium [5]. Lesions are restricted to the large intestine and are characterized by superficial necrosis of the mucosa associated with mucofibrinous hemorrhagic exudate. Microscopic lesions are hyperplasia of goblet cells, superficial necrosis, exudation of fibrin and blood with neutrophilic inflammatory infiltration resulting in dehydration and death in untreated animals [5,6,7]. 

Porcine proliferative enteropathy (PPE) is caused by *Lawsonia intracellularis*, a microaerophilic and obligatory intracellular bacterium responsible for weight loss and increased feed conversion [8,9]. This disease has three forms: acute, chronic, and subclinical [9,10]. The chronic form affects growing animals between 6 and 20 weeks of age and is characterized by anorexia and pasty diarrhea. The acute form is characterized by hemorrhagic enteritis with sudden death [9] in finishing pigs and gilts [10,11]. The subclinical form is the most frequent [12] with the intermittent shedding of *L. intracellularis* associated with performance loss [13]. The gross lesions are observed in the small and large intestine and consist of thickened and corrugated mucosa and, in chronic cases, there is necrosis, with deposits of fibrin and cellular debris [11]. The microscopic lesions are characterized by hyperplasia of crypt immature enterocytes and lack goblet cells [9].

The pathogeneses of both diseases are complex and poorly understood [14,15]. Some studies have shown that the presence of clinical signs is strongly influenced by diet and microbiota [16,17]. Several studies have demonstrated the effects of different diets on colonization and the presence of microorganisms that are important for clinical sign manifestation [18,19,20]. Thus, it is possible to control some bacterial populations by manipulating gastrointestinal microenvironmental conditions. Some studies have shown that gnotobiotic pigs experimentally infected with *B. hyodysenteriae* or *L. intracellularis* do not develop typical swine dysentery or proliferative enteropathy, respectively, demonstrating the importance of intestinal microbiota in disease development [18,19,21,22,23,24].

*L. intracellularis* and *B. hyodysenteriae* coinfection have been observed in the field, and these cases are increasing in the diagnostic routine. Few reports have demonstrated coinfection between these two agents [25,26,27,28]. The possible interaction between these two agents could be related to the microenvironment generated by the microbiota, which triggers the associated disease.

The aim of this study was to evaluate the infection conditions, possible synergism, and fecal microbiota profiles of animals coinfected with *B. hyodysenteriae* and *L. intracellularis*.

## 2. Materials and Methods

### 2.1. Animals and Experiment

This study was approved by the Ethics Committee on Animal Experimentation of Universidade Federal de Minas Gerais (CEUA #157/2016). 

Forty-five four-week-old piglets from a commercial farm with no history of *Brachyspira* spp., *Lawsonia intracellularis*, or *Salmonella* spp. were randomly divided into four groups: negative control (NEG) with 11 animals, coinfection with *B. hyodysenteriae* and *L. intracellularis* (CO) with 12 animals, *B. hyodysenteriae* (BRA) with 11 animals, and *L. intracellularis* (LAW) with 11 animals. 

All piglets were housed in the experimental facility and maintained for 14 days to acclimatize to food and water ad libitum. Throughout the experimental period, the feed was free of antimicrobials and other compounds that could interfere with the experiment. Each experimental group was housed in a different room and handled by different personnel to maintain strict biosecurity and avoid cross-contamination. No therapeutic or metaphylactic treatments were administered.

### 2.2. Inoculum Preparation 

#### 2.2.1. Lawsonia Intracellularis

A porcine isolate of *L. intracellularis* (BRPHE01_E5) obtained from an animal with proliferative hemorrhagic enteritis was cultured in multiple passages (up to 12 and 23 in vitro) in cell culture using McCoy cells (ATCC CRL 1696) in DMEM (Dulbecco’s modified Eagle’s—JRH Lenexa, KS, USA), supplemented with 5% fetal bovine serum (FBS) and 1% L-glutamine and incubated for five days in a gas concentration of 8% O_2_, 8.8% CO_2_ and 83.2% N_2_ at 37 °C, as described by Guedes and Gebhart [29]. 

Bacteria were suspended in a sucrose phosphate glutamate (SPG) solution to prepare the inoculum with 10% (FBS). On day zero, animals of the LAW and CO groups were inoculated intragastrically using an intragastric feeding tube with 50 mL of pure culture of *L. intracellularis* containing 2.76 × 10^6^ bacteria/mL. Animals in the NEG and BRA groups received a sterile SPG solution via the same route on the same day.

#### 2.2.2. Brachyspira Hyodysenteriae

The *B. hyodysenteriae* strain used as an inoculum was isolated from a growth-finishing pig with severe swine dysentery in 2013 from a Brazilian hog farm. It was cultivated on Trypticase Soy Agar (TSA) supplemented with 5% sheep blood and 12.5 mg/L rifampicin, 200 mg/L spectinomycin, 50 mg/L vancomycin, and 12.5 mg/L of colistin [30], under an anaerobic atmosphere (80% N_2_, 10% CO_2_ and 10% H_2_), at 37 °C for three days. 

After growth on solid medium, the agar plates were washed with sterile PBS and incubated in trypticase soy growth broth (TSB), enriched with 0.5% glucose, 0.2% NaHCO_3_, 0.05% L-cysteine-HCl, 1.0% yeast extract, 10% FBS and 5% porcine fecal extract [31] in a proportion of 1:100 mL (wash: broth) for 21 h at 37 °C in a shaker incubator, followed by inoculation of the animals. Pigs from the BRA and CO groups were intragastrically inoculated for three consecutive days at 7, 8, and 9 days post infection (dpi) with 50 mL of inoculum containing 5.31 × 10^6^ bacteria/mL, as described by Jacobson et al. [14] and Rubin et al. [32]. The NEG and LAW groups received 50 mL of sterile TBS via the same route on the same day. 

A low challenge dose for both microorganisms was used to maximize the chances of observing a synergic effect in coinfected animals, as a high dose would probably aggravate clinical conditions and mask them.

### 2.3. Clinical Evaluation

After inoculation, all animals were evaluated daily for clinical signs, mainly fecal consistency, based on the following scores: 0 = normal, 1 = semi-solid consistency, 2 = liquid, and 3 = severely watery liquid with the addition of 0.5 for the presence of mucus and/or blood [30]. Stool samples were collected on days −5, 3, 6, 10, 12, 15, 18, and 21, and tested by qPCR for *B. hyodysenteriae* and *L. intracellularis*, and bacterial isolation was performed for *B. hyodysenteriae* as described above.

### 2.4. Quantitative PCR

Feces DNA was extracted using a commercial QIAamp DNA Stool Mini Kit (Qiagen Inc., Toronto, ON, Canada) according to the manufacturer’s instructions. For *B. hyodysenteriae*, the primers JH0073 (5′-AGT GAA ATA GTT GCT CAT ATC AAA-3′) and the JH0074 (5′-GCA TCA CTG ATT AAA GAA CCA ATT-3′)-targeting *Nox* gene were used to perform qPCR according to Rubin et al. [32].

For *L. intracellularis*, primers bcL.intra114f (5′-CACTTGCAAACAATAAACTTGGTCTTC-3′) and bcL.intra-263r (5′-CATTCATATTTGTA-CTTGTCCCTGCA-3′) associated with the intra201p probe (TCCTTGATCAATTTGTTGTGGATT-GTATTCAAGG) were used to performed TaqMan qPCR and PCR Mastermix (TaqMan Universal PCR Mastermix; Applied Biosystems, Waltham, MA, USA), according to the manufacturer’s instructions. All reactions were performed in duplicate, and each reaction included the standard curve and negative control, being analyzed in the QuantStudioTM Real Time PCR v1.2 software. The standard curve was performed through serial dilutions of a known concentration of bacteria per mL for both infectious agents. For each reaction, the standard curve was included and the correlation between the cycle threshold (CT) and the known concentration of the curve was performed [33].

### 2.5. Pathology

All pigs were euthanized and necropsied at 21 dpi. Clinically debilitated animals, according to the CEUA criteria, were euthanized and evaluated post-mortem throughout the experiment. Macroscopic lesions were evaluated via necropsy, and fragments of the small intestine, large intestine, and mesenteric lymph nodes were collected and fixed in 10% formalin for histopathological examination. Samples, including rectal feces and mucosa scraped from the ileum, cecum, and colon, were collected for the detection of other agents.

### 2.6. Bacterial Isolation

Feces and large-intestine scraping were cultivated with selective medium on TSA Tryptic Soy Agar (Tryptic Soy Agar, DIFCO, Taiwan, China, cat no. 211043) supplemented with 5% sheep blood, 6.25 mg/μL rifampicin (Rifampicin, Sigma-Aldrich, St. Louis, MO, USA, cat no. R3501), and 800 mg/L spectinomycin (Sigma-Aldrich, cat no. S9007), 25 mg/μL vancomycin (Vancomycin, Sigma-Aldrich, cat no. V2002), 25 mg/L of colistin (Colistin, Sigma-Aldrich, cat no. C1511) [32], and incubated for at least three days at 42 °C in jars with an anaerobic atmosphere. Anaerobic conditions were generated using a vacuum pump filled with a mixture of N_2_ (80%), CO_2_ (10%), and H_2_ (10%) gases. To obtain pure colonies, several passages were performed until isolation. The isolates were stored in a freezer at −80 °C.

For differential diagnosis, at the end of the experiment, mucosal scrapings from the small intestine were cultivated on blood agar and MacConkey to evaluate the growth of enterotoxigenic *Escherichia coli* and scrapings from the large intestine mucosa were cultivated in Rappaport broth and Hectoein agar for *Salmonella* spp.

### 2.7. Histopathology

Two-centimeter fragments from the jejunum, ileum, cecum, and spiral colon were collected, fixed in 10% buffered formalin, processed, and stained with hematoxylin and eosin [34]. Microscopic lesions associated with *B. hyodysenteriae* and *L. intracellularis* infections were recorded. Cecum and colon sections were evaluated according to the intensity and distribution of the following lesions: superficial necrosis, hemorrhage, enterocyte hyperplasia, goblet cell hyperplasia (IG), crypt abscesses, and lamina propria neutrophil infiltration. All parameters were individually classified as follows: 0, absent; 1, mild; 2, moderate; and 3, severe. The final score was determined as a “*composite score*”. Two blinded pathologists evaluated all histological sections, and the mean of the two counts was used for all analyses.

### 2.8. Immunohistochemistry

Histological sections were stained using immunohistochemistry (IHC) with labeled streptavidin and rabbit polyclonal antibodies specific to *L. intracellularis* [35]. The IHC staining was graded from 0 to 4, with 0 indicating no staining, grade 1 when focused antigenic staining was observed in crypts or lamina propria, grade 2 for multiple foci of antigenic staining (about 25% of the crypts), grade 3 for part of the mucosa with positive staining (26–75% of the crypts), and grade 4 for more than 75% of crypts with antigen labeling.

### 2.9. Intestinal Microbioma

#### 2.9.1. Samples

Rectal feces collected from all animals on days −5 and 21 were used to assess the fecal microbiome. Until the date of processing, the samples were stored at −80 °C.

#### 2.9.2. 16S Sequencing

Total DNA was extracted from 200 mg of the fecal sample using a commercial kit (QIAmp DNA from Fecal Mini Kit, Qiagen Inc., Toronto, ON, Canada) following the manufacturer’s recommendations. After extraction, the DNA was quantified using a Qubit^®^ 2.0 Fluorometer and Qubit^®^ dsDNA Hs Assay Kit (Life Technologies, Carlsbad, CA, USA).

For genomic sequencing, the “*Fusion”* method developed for microbiome analysis using next-generation sequencing by the Ion Torrent 16S Metagenomics kit that amplifies the V4 hypervariable region of the 16S rRNA gene was performed. Primers were customized with one reverse primer and 96 forward primers with barcodes. The hypervariable V4 16S rRNA gene was amplified using the fusion primers [36,37]. 

For the initial PCR reaction, 1 × Platinum^®^ PCR SuperMix High Fidelity was used with 5 µM of each fusion primer, 20–50 ng genomic DNA, and sterile deionized water. The cycles used were 1 cycle of initial denaturation at 94 °C for 3 min, followed by 40 cycles of denaturation at 94 °C for 30 s, annealing at 58 °C for 30 s, and extension at 68 °C for 1 min/kb. The PCR products were confirmed using a QiAxcel Advanced System (Qiagen), purified with Agencourt^®^ AMPure XP Reagent (Beckman Coulter), quantified using a Qubit^®^ 2.0 Fluorometer and Qubit^®^ dsDNA HS Assay Kit (Life Technologies), and readjusted to a final concentration of 26 pM.

Emulsion PCR was performed by emulsion breaking and enrichment using OTON OT-Q™ Ion PGM™ kit (# A29900), according to the manufacturer’s instructions. The sample was prepared for sequencing using the Ion PGM™ Hi-Q™ View Sequencing Kit (#A30044). To determine the quality, two synthetic 16S microbial communities (Mock Communities) of species with known genomes were used. The first community, HMD-782D, contained 20 bacterial strains with 100,000 copies per organism per microliter. The second community, HMD-783D, contained 20 bacterial strains ranging from 1000 to 1,000,000 copies per organism per microliter. Both reagents were obtained from BEI Resources, NIAID, and NIH as part of the Human Microbiome Project: Genomic DNA from Microbial Mock Community B (Even, Low Concentration), v5.1L, for 16S RNA gene sequencing, HM-782D and Genomic DNA of Microbial Mock Community B (Staggered, Low Concentration), v5.2L, for 16S rRNA Gene Sequencing, HM-783D.

### 2.10. Bioinformatics Analysis

#### 2.10.1. Metagenomic Classification Pipeline

Raw data sequences were processed following the operational taxonomic unit classification pipeline of the Brazilian Microbiome Project [38]. In the initial step, the raw data underwent filtering using a custom script (available at: https://github.com/aquacen/fast_sample) with the following parameters: “*-n 100*” (for all reads), “*-s 160*” (including only reads with a length of at least 160 bp), “*-b 310*” (trimming reads with a length of at least 310 bp), “*-l 0*” (no left clip), and “*-q 20*” (trimming 3′ end reads with a Phred quality score < 20). 

The subsequent step involved the utilization of Uparse software [39] for read labeling and Usearch (usage version 10.0.240) [40] for quality filtering (-fastq_filter -fastq_maxee 0.8), removal of replicated reads (-fastx_uniques -sizeout), sorting by size (-sortbysize -minsize 2), grouping into OTUs (-cluster_otus), and mapping the raw data onto OTUs (-usearch_global -strand plus -id 0.97). Uparse was employed to generate the list of OTUs and convert the files into an OTU table, while the initial version of QIIME software [41] was utilized for taxonomy assignment (similarity 0.7), alignment of OTU sequences, and construction of the phylogenetic tree.

Finally, Biom software version 2.1.5 [42] was employed to perform the following tasks: convert the Biom table, add taxonomy metadata in QIIME (--observation-header OTUID, taxonomy, confidence --sc-separated taxonomy --float-fields confidence), and organize the OTU table. The Usearch program, version 10.0.240, included chimera filters in the OTU clustering step (cluster_otus). The two “barcodes” containing “Mock” communities underwent the same processing steps.

#### 2.10.2. Alpha and Beta Diversity Analysis

QIIME software was used to filter samples from the OTU table (-n 1000). Unique rarefaction (-d 1000) was performed to assess beta diversity, create a weight (--metrics unweighted_unifrac) and weight (-metrics weighted_unifrac) beta diversity, and multiple rarefactions (-m 10 -× 50,000 -s 2000) were used for alpha diversity. 

### 2.11. Statistical analysis

The Kruskal–Wallis test was performed to compare the diarrhea scores, qPCR results, histopathological lesions, and immunohistochemistry results. Fisher’s exact test was used to compare the bacterial isolation results. The R software package ggplot2 was used to generate α and β diversity plots with QIIME results and the heat map and statistical comparisons of the familiar abundance using the Pairwise Wilcox Test with Bonferroni correction. Only households with a representative ≥1% in nearly one barcode were included in the analysis [43]. Results were considered significant with *p* ≤ 0.05. 

## 3. Results

### 3.1. Clinical Signs

At day 5, all animals tested negative for *L. intracellularis* and *B. hyodysenteriae* in the stool samples by qPCR. Pigs from CO group had watery diarrhea with a score ≥ 3 for the first time on day 12 post infection. Two animals from the same group had a diarrhea score of 3 for two consecutive days, from 4 and 6 dpi; however, the diarrhea stopped thereafter. One of the two animals with diarrhea at 6 dpi was PCR positive for *L. intracellularis*. At the end of the experiment, nine animals in this group had diarrhea. In the BRA group, the pigs developed diarrhea on day 14, affecting five animals. In the LAW group, four pigs had diarrhea; however, only two animals had more persistent diarrhea, while the others had diarrhea for just two days. In the NEG group, two animals had a diarrhea score of 3; however, one animal had diarrhea for only one day, and the others had diarrhea for two days. Comparison of fecal scores revealed statistically significant differences between the CO group and all other groups after 18 dpi, as shown in Figure 1.

Due to poor clinical conditions, two animals from the CO group (17 and 18 dpi) and one from the BRA group (16 dpi) died 3 days after starting bloody diarrhea. The animals exhibited severe clinical signs of swine dysentery. Necropsy revealed severe fibrinonecrosis and catarrhal colitis associated with mesocolonic edema and hyperemia. One animal from the LAW group was euthanized at 18 dpi, which had a low clinical score (0–1) and hyperplasia of enterocytes, and a positive staining score of 2 by immunohistochemistry for *L. intracellularis* was observed. 

### 3.2. Anatomopathological Findings

#### 3.2.1. Gross Lesions

Macroscopic lesions were detected in the large intestines of animals inoculated with *B. hyodysenteriae*, CO (10/12), and BRA (4/11). In the CO group, the lesions in the large intestine were characterized by moderate diffuse mucohemorrhagic colitis and severe diffuse fibrin necrohemorrhagic catarrhal colitis. In the BRA group, lesions were observed in four animals ranging from multifocal moderate mucohemorrhagic colitis to severe diffuse fibrin hemorrhagic colitis in four animals. In the LAW group, only two animals had moderate hyperemia in the ileum. In the NEG group, one animal presented with moderate hyperemia of the colonic serosa. The most important lesions are shown in Figure 2. 

#### 3.2.2. Microscopic Lesions

Histological findings are presented in Figure 3. In the large intestine, lesions were more severe in the CO group for all parameters except for enterocyte hyperplasia when compared to the NEG and LAW groups (*p* < 0.05). The BRA group differed from the CO group only in goblet cell hyperplasia. The LAW group had significantly more crypt abscesses than the NEG group. No significant changes were observed in the small intestines or lymph nodes.

### 3.3. Immunohistochemistry

Mild immunolabelling of *L. intracellularis* in animals in the CO and LAW groups differed from that of the BRA and NEG groups (Figure 4 and Figure 5), which showed no antigen labelling. Five animals from the CO group and four from the LAW group were IHC-positive for *L. intracellularis*. 

### 3.4. Bacterial Isolation

No enterotoxigenic *Salmonella* spp. or *Escherichia coli* growth was observed in stool samples collected from any of the pigs before inoculation. *B. hyodysenteriae* was isolated from different fragments of the intestine and feces of 11 of the 12 pigs in the CO group. In the BRA group, *B. hyodysenteriae* was isolated from 5 of 11 pigs. *B. hyodysenteriae* was not isolated from any pigs in the LAW or NEG groups (Figure 6). 

### 3.5. Quantitative PCR

The first positive qPCRs for *L. intracellularis* in the CO and LAW groups were detected at 3 dpi (Figure 7). On the last day of evaluation (21 dpi), all animals in both groups tested positive for *L. intracellularis*. qPCR for *B. hyodysenteriae* in the CO group was detected in 10 of the 12 positive animals 10 dpi (3 days after *B. hyodysenteriae* inoculation). In the BRA group, 7 out of 11 pigs were qPCR-positive, starting shedding at 10 dpi and 3 days after inoculation with *B. hyodysenteriae.*

### 3.6. Microbiome Analysis

Ion Torrent sequencing obtained 79% chip coverage (~8.5 mi reads), and 40% of the reads were removed due to polyclonality (~5 mi after filtering). At the end of the filtering, 2,982,952 reads were analyzed, with an average of 33,143 reads per sample. Regarding MOCK communities, 4,631,158 reads and 1,648,206 low-quality reads (Phred < 20) were removed, leaving 2,982,952 reads with an average of 33,143 reads per sample.

There was no significant difference in richness among the groups; however, it was possible to observe greater diversity in the NEG group than in the other groups, and less diversity in CO compared to the other groups at 21 dpi. The results of the differential microbiome abundances among the groups are shown in Appendix A.

Higher relative abundances of the genera *Prevotella*, *Anaerovibrio*, *Bacteroides*, *Butyricimonas*, *Desulfovibrio*, *Fusobacterium*, and *p75-a5* were observed in the CO group than in the other groups (*p* < 0.05). The BRA group had a higher frequency of members of the *Clostridium* genus. Animals in the LAW group showed increased numbers of *Megasphaera* and *Dialister* (*p* < 0.05). The NEG had a higher frequency of *Odoribacter*. The relative abundances are shown in Figure 8, Figure 9, Figure 10 and Figure 11. There was no statistically significant difference in beta diversity among the groups; however, an evident dispersion was observed between the CO and NEG groups at 21 dpi (Appendix A). The beta diversity results are presented in Appendix A.

## 4. Discussion

*L. intracellularis* and *B. hyodysenteriae* coinfection has been reported in diagnosed field cases [26,27,28]; however, no studies have evaluated coinfection in experimentally infected animals considering anatomopathological evaluation and the fecal microbiome.

In the present study, more severe clinical signs and macroscopic and microscopic lesions were observed in animals in the CO group than in those in the other groups. Higher diarrhea scores and fecal shedding of *B. hyodysenteriae* were detected in the CO group at 10 dpi, and more animals in this group tested positive for bacterial isolation. In the BRA group, fewer animals were affected by swine dysentery; however, the affected animals had a typical disease presentation [5,32,44].

In the LAW group, no expressive diarrhea was observed; however, moderate microscopic lesions of proliferative enteropathy, positive staining for *L. intracellularis* by immunohistochemistry, and positive qPCR results were observed in both the LAW and CO groups. Therefore, *L. intracellularis*-infected animals develop subclinical forms of proliferative enteropathy. A possible explanation could be the inoculum concentration of *L. intracellularis* (10^6^), which is considered low based on the estimated infective dose of 10^3^ [45] and compared to studies in which there was a typical clinical manifestation (10^8^–10^10^) [29,46,47]. In the present study, the inoculum used was a pure culture of *L. intracellularis* instead of scraping the mucosa. This type of inoculum was preferred as the fecal microbiome would be evaluated, thus preventing the presence of microorganisms other *than L. intracellularis*. Clinical signs and lesions were more severe in studies using pure cultures compared to mucosal scrapings [8,48,49].

Based on the anatomopathological findings, qPCR results, isolation, and immunohistochemistry, it was possible to confirm the success of the experimental challenge model, with significant differences in all variables, particularly more pronounced in animals from the CO group, followed by the BRA group.

Our results suggest that the presence of *L. intracellularis* in the CO group significantly increases the severity of swine dysentery. We hypothesized that *L. intracellularis* infection induces early changes that allow for the more effective colonization and establishment of *B. hyodysenteriae* in the large intestine or impair the host intestinal immune response, facilitating higher colonization of *B. hyodysenteriae*. Furthermore, an important mechanism in the pathogenesis of proliferative enteropathy is the interference with intestinal epithelial cell proliferation by increasing the proportion of immature enterocytes [50,51]. Injury caused by cell disruption associated with necrosis and the presence of immature enterocytes may predispose *Brachyspira* spp. to infection. Another potential factor is the immunosuppressive mechanism induced by *L. intracellularis* infection, with limited infiltration of inflammatory cells during the development of proliferative lesions [9,52,53]. MacIntyre et al. [52] characterized the immune response associated with *L. intracellularis* infection and demonstrated an association between the peak of infection and a reduced number of T and B lymphocytes with the downregulation of the adaptive immune response. The high colonization capacity of *B. hyodysenteriae* in the large-intestine environment may have allowed a high rate of colonization and expression of swine dysentery.

Intestinal coinfections in growing–finishing pigs are considered common in the diagnostic routine and are more serious because they have two or more pathogens that can synergistically colonize the same site, leading to more expressive lesions, as commonly observed in polymicrobial diseases [54,55,56]. In pigs, coinfection is observed in cases of enzootic pneumonia, atrophic rhinitis, and circovirosis [57]. In a study on *Mycoplasma hyopneumoniae*, more severe lesions were observed when associated with PCV2 [58]. An association of circoviruses with *Salmonella* spp. and influenza A virus has been observed, with greater susceptibility observed when in association with one another [59,60]. The inoculation of porcine epidemic diarrhea virus (PEDV) and *Clostridium perfringens* type A enhanced disease severity in weaned pigs compared to a single infection, characterized by higher viral fecal shedding and more severe villous atrophy in the small intestine [61]. *Trichuris suis* and *L. intracellularis* coinfection have demonstrated a synergistic effect that induces severe necrotizing proliferative colitis and a reduction in the growth rate in pigs [62].

Studies evaluating the microbiome of animals inoculated with *B. hyodysenteriae* or *L. intracellularis* have been previously described [20,63,64]. Regarding the intestinal microbiome, a decrease in microbial richness was observed at the end of the study period in the CO group, which was less pronounced in the BRA and LAW groups, a factor commonly observed in cases of dysbiosis. The NEG group showed a less significant decrease in diversity than the other groups. There is increasing evidence that dysbiosis of the intestinal microbiota is associated with the pathogenesis of intestinal and extra-intestinal disorders in humans. In these cases, the mechanisms leading to disease development involve a relationship between the microbiota and its associated metabolic products in the host’s immune system [65]. For example, significant microbiota changes in patients with Chron’s have been associated with dysbiosis [66,67].

Studies on microbial relationships are essential for understanding the pathogenesis of diseases with complex mechanisms of colonization and infection progression, such as swine dysentery and proliferative enteropathy. In the present study, it was possible to observe greater relative abundance with a statistical difference in CO compared to the other groups for the genera *Prevotella*, *Anaerovibrio*, *Bacteroides*, *Butyricimonas*, *Desulfovibrio, Fusobacterium* and *p75-a5*, and a greater abundance of *Clostridium* in the BRA group. In the LAW group, *Megasphaera*, *Dialister*, and *Odoribacter* were significantly more frequent than in the NEG group (*p* < 0.05).

*Prevotella* is abundant in the gastrointestinal tract of pigs [68,69]. Some studies found it was significantly abundant in mucosal scrapings from animals with swine dysentery and pigs positive for *Salmonella enterica* [61,62]. Recently, it has been associated with irritable bowel syndrome in humans and mice [70]. An interesting factor related to this genus is the reduction in *Prevotella* spp. in the colon microbiota when chicory is included in the diet [71]. Studies testing chicory-based nutraceuticals showed a decrease in swine dysentery cases [72].

The genus *Anaerovibrio* has not yet been clearly elucidated but is commonly found in sheep and cattle. Microorganisms of this genus have a high capacity to hydrolyze lipids [73], a characteristic that is probably important for colonization by *B. hyodysenteriae*, as bacteria of this genus require compounds such as cholesterol for their metabolism [74,75]. *Bacteroides*, *Fusobacterium*, and *Clostridium* are important genera for *B. hyodysenteriae* infection [19] and were observed in the CO group. A significant increase in *Fusobacterium* was also reported in pigs with porcine epidemic diarrhea virus (PED) [76] and in animals with nonspecific diarrhea [77]. The genus *Clostridium* has several species that have both beneficial and negative potential as hosts [78]. This genus in expressive numbers has been observed in pigs with necrotizing enterocolitis [79].

The increase in *Desulfovibrio* was also demonstrated by Burrough et al. [63] in swine-dysentery-diseased animals. *Desulfovibrio* is also consistently increased in humans with ulcerative colitis [80]. They are sulfate-reducing bacteria capable of degrading mucins, decreasing the mucosal barrier [81], and facilitating *B. hyodysenteriae* adherence to the intestinal mucosa [82]. The genera *p-75-a5*, *Butrycimonas, Odoribacter,* and *Megasphaera* have been poorly reported. The *p-75-a5* genus has been observed in the feces of pre-weaning pigs [83] and wild birds [84]. Recently, it was revealed that there was an increased relative abundance of *p-75-a5* in zinc oxide- and antimicrobial-treated groups of pigs [85]. The *Butyricimonas* genus has demonstrated only two cases of human infection with *B. virosa* in patients with irritable bowel syndrome [86]. More abundant in the NEG group in the present study, the genus *Odoribacter* was more abundant in healthy humans than in individuals with intestinal mucosal inflammation [87] and is considered an important genus for intestinal homeostasis [88]. A greater abundance of *Megasphaera* and *Dialister* was observed in animals in the LAW group. *Megasphaera* is considered an inhibitor of *B. hyodysenteriae* colonization [72]. *Dialister* are obligatorily anaerobic or microaerophilic bacteria. They are found in the oral cavity of healthy humans, oral infections, blood cultures, and abscesses of the brain, nasopharynx, and mouth [89].

## 5. Conclusions

Animals co-infected with *B. hyodysenteriae* and *L. intracellularis* showed more severe clinical signs as well as gross and histological lesions than animals infected with only one of these agents. We hypothesized that *L. intracellularis* is an immunosuppressive factor that favors *B. hyodysenteriae* colonization. Regarding the fecal microbiota, all groups showed significant differences in the relative abundance of multiple species. We observed a greater relative abundance of *Prevotella*, *Anaerovibrio*, *Bacteroides*, *Butyricimonas*, *Desulfovibrio*, *Fusobacterium,* and *p75-a5* in the CO group than in the other groups and a greater abundance of *Clostridium* in the BRA group than in the other groups. In the LAW group, *Megasphaera*, *Dialister*, and *Odoribacter* were more abundant than in the NEG group.

## Figures and Tables

**Figure 1 animals-13-02611-f001:**
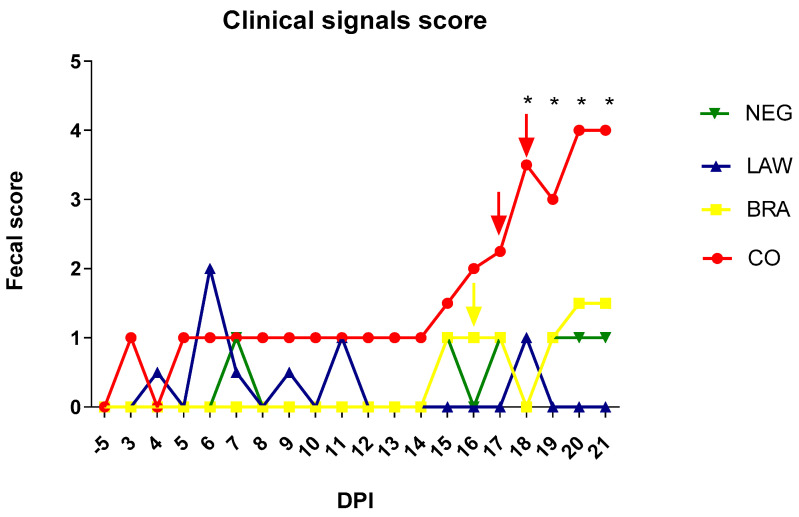
Mean of fecal score between −5 to 21 dpi in coinfection (CO) in *B. hyodysenteriae* (BRA), *L. intracellularis* (LAW), and Negative (NEG) groups. (*) *p* ≤ 0.05. 0 = normal, 1 = semi-solid consistency, 2 = liquid and 3 = severely watery liquid, with addition of 0.5 for presence of mucus and/or blood [30]. Colored arrows indicate mortality of animals per group.

**Figure 2 animals-13-02611-f002:**
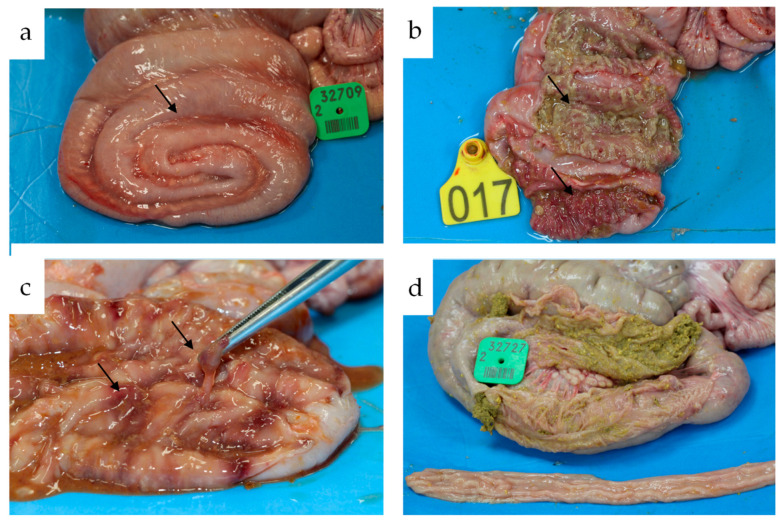
Macroscopic lesions observed at necropsy. (**a**) Animal from CO group with marked mesocolon edema (arrows). (**b**) Animal from CO group with severe diffuse fibrin necro hemorrhagic catarrhal colitis (arrows). (**c**) Animal from the BRA group with excess mucus (tip of the forceps) associated with multifocal mucosal hemorrhage (arrows). (**d**) Animal from the NEG group with macroscopic normal colon.

**Figure 3 animals-13-02611-f003:**
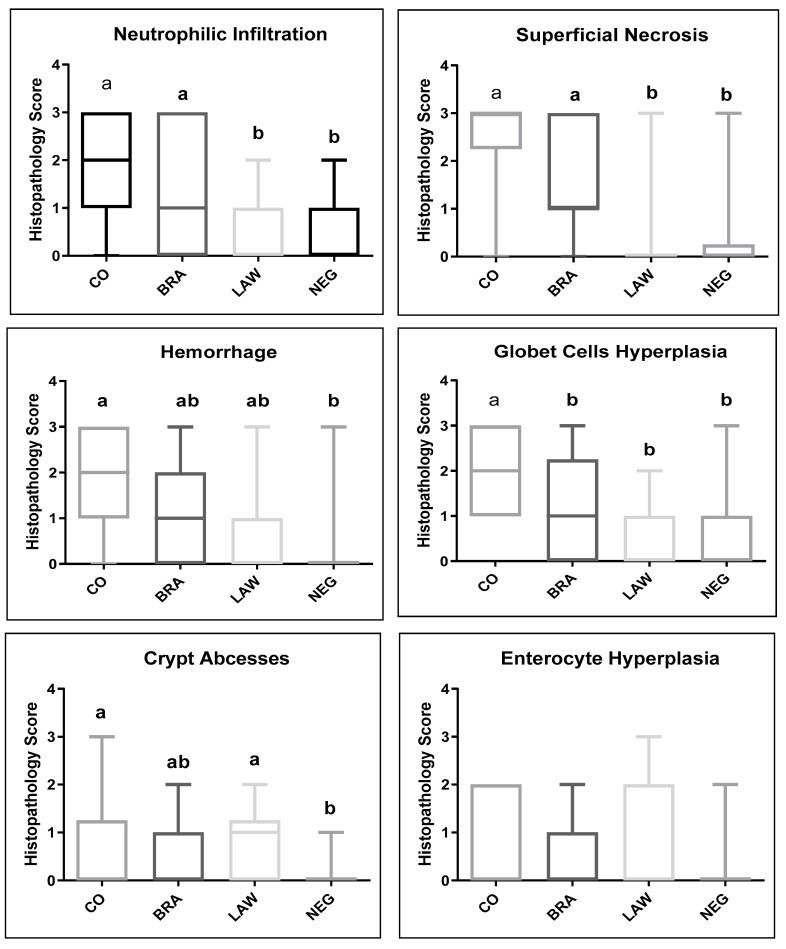
Median of histopathological scores at large intestine in coinfection (CO) in *B. hyodysenteriae* (BRA), *L. intracellularis* (LAW) and Negative (NEG) groups. Cecum and colon sections were evaluated according to the intensity and distribution of the following lesions: superficial necrosis, hemorrhage, enterocyte hyperplasia, goblet cell hyperplasia (IG), crypt abscesses, and lamina propria neutrophil infiltration. All parameters were individually classified as follows: 0, absent; 1, mild; 2, moderate; 3, severe. Different letters indicate statistical differences (*p* ≤ 0.05; Kruskal–Wallis test) among groups.

**Figure 4 animals-13-02611-f004:**
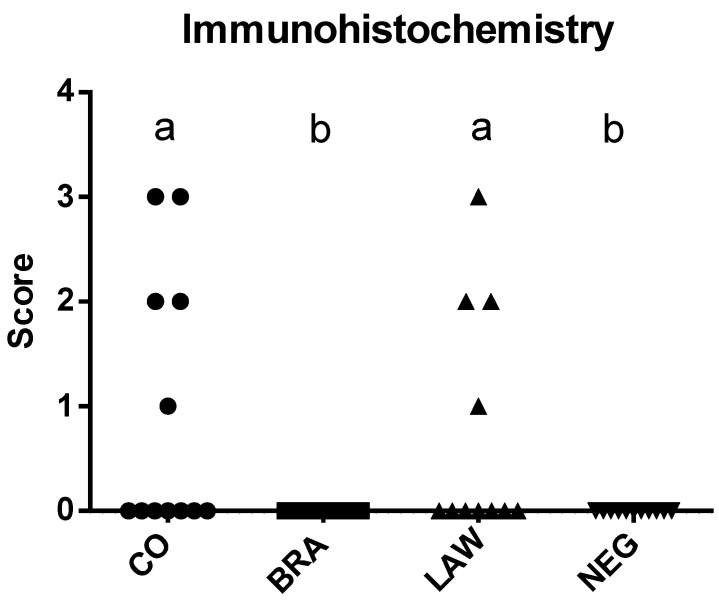
Median immunohistochemistry scores for *L. intracellularis* per group. The immunohistochemistry staining was graded from 0 to 4, with 0 indicating no staining, grade 1 when focal antigenic staining was observed in crypts or lamina propria, grade 2 for multiple foci of antigenic staining (about 25% of the crypts), grade 3 when 26–75% of the crypts in the mucosa had positive staining (26–75% of the crypts), and grade 4 for more than 75% of crypts with antigen labeling. Different letters indicate statistical differences (*p* ≤ 0.05; Kruskal–Wallis test) among groups.

**Figure 5 animals-13-02611-f005:**
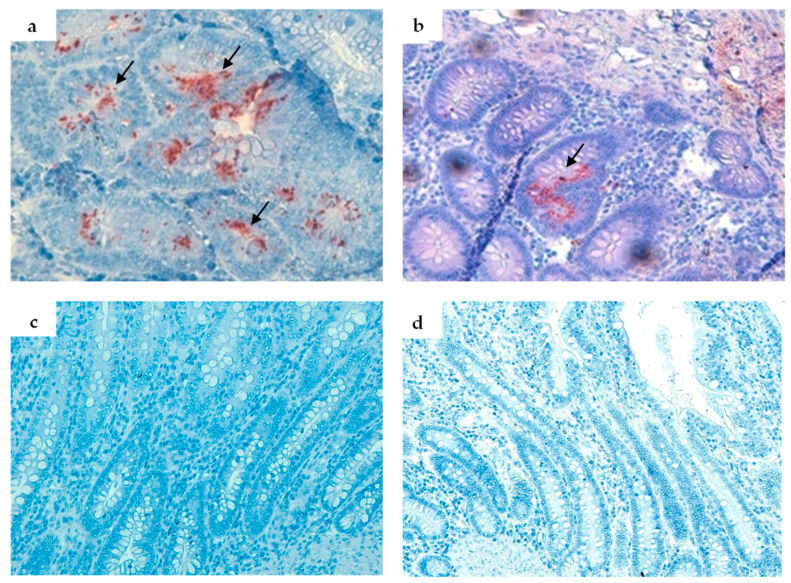
Immunohistochemistry of pigs experimentally infected with *L. intracellularis* at 21 dpi. (**a**) Group (CO)—Immunostaining of *L. intracellularis* (in red, pointed by black arrows) in crypt enterocytes (grade 4) (40×). (**b**) (LAW) Immunostaining of *L. intracellularis* (in red, pointed by arrows) in crypt enterocytes in the ileum lamina propria (grade 1) (20×). (**c**) Animals from BRA (20×). (**d**) NEG groups with negative immunostaining (20×). AEC (3-Amino-9 Ethylcarbazole).

**Figure 6 animals-13-02611-f006:**
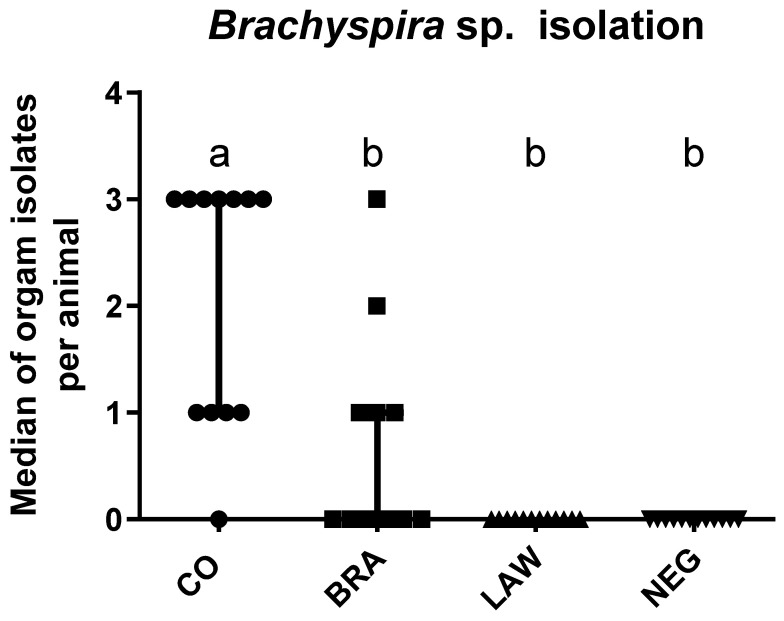
Median of Brachyspira sp. isolation from feces and large intestine scraping by group. Different letters indicate statistical differences (*p* ≤ 0.05; Kruskal–Wallis test) among groups.

**Figure 7 animals-13-02611-f007:**
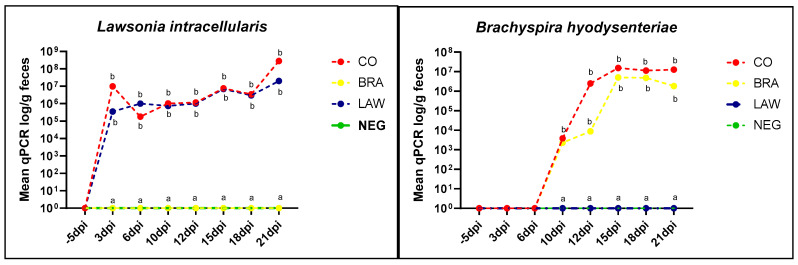
Mean of qPCR for Lawsonia intracellularis and Brachyspira hyodysenteriae during the experiment. Quantitative PCR using standard curve. There is no difference of shedding between *Brachyspira hyodysenteriae*- and *Lawsonia intracellularis*-inoculated groups. Different letters indicate statistical differences (*p* ≤ 0.05; Kruskal–Wallis test) among groups at different days post-infection (dpi).

**Figure 8 animals-13-02611-f008:**
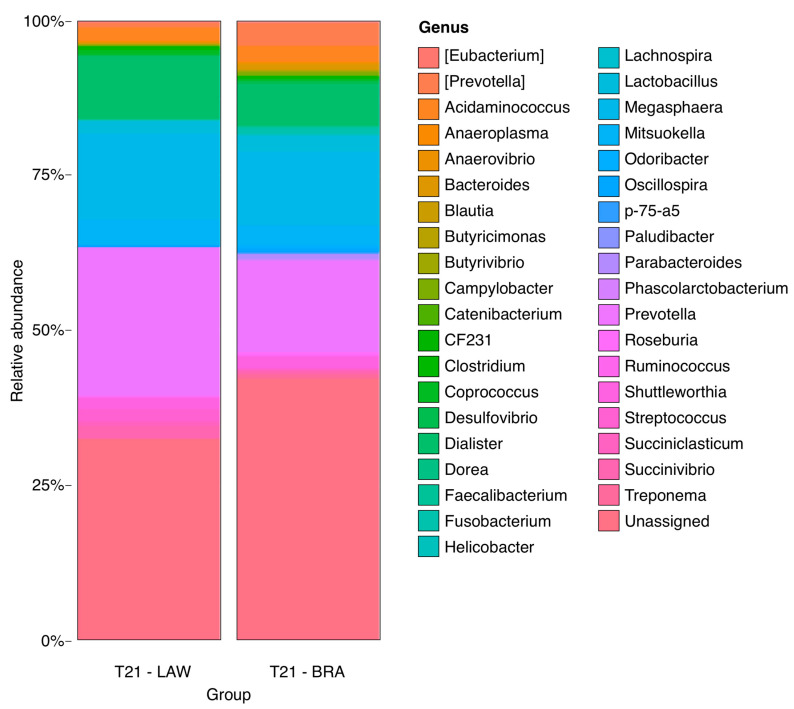
Bar graphs showing proportional abundance of major genus at LAW group on the left and BRA group on the right at 21 dpi. A higher relative abundance of *Parabacteroides* (*p* = 0.012886830493507) at BRA and *Streptococcus* (*p* = 0.012886830493507) in the LAW group.

**Figure 9 animals-13-02611-f009:**
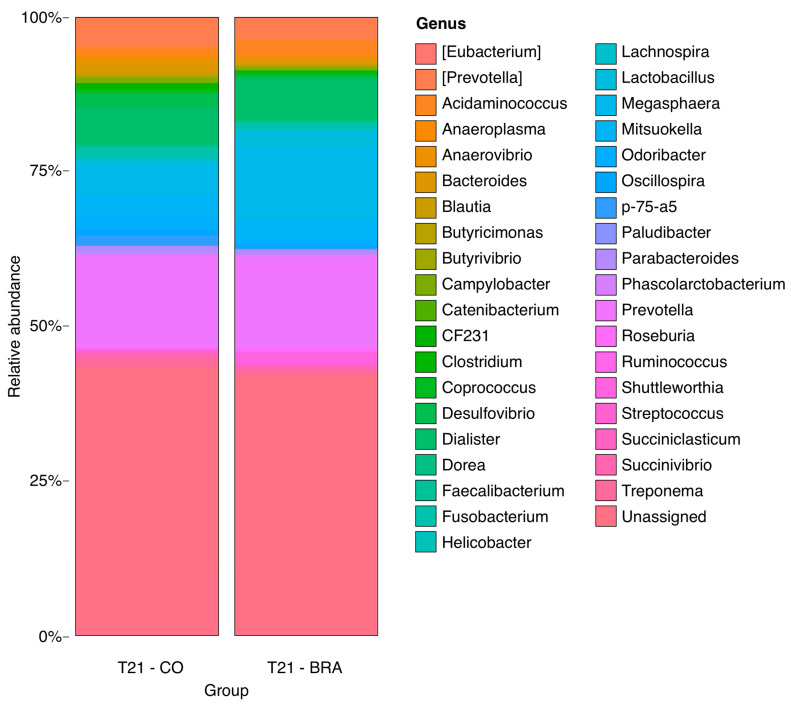
Bar graphs showing proportional abundance of main genus in CO animals on the left and BRA on the right at 21 dpi. A higher relative abundance of *Prevotella* (*p* = 0.012879064669346), *Fusobacterium* (*p* = 0.0355240368313795), *Lactobacillus* (*p* = 0.0149386031285565), *p-75-a5* (*p* = 0.0474483843880802) was observed in animals in the CO group and, in the BRA group, greater abundance of *Clostridium* (*p* = 0.010705311873033).

**Figure 10 animals-13-02611-f010:**
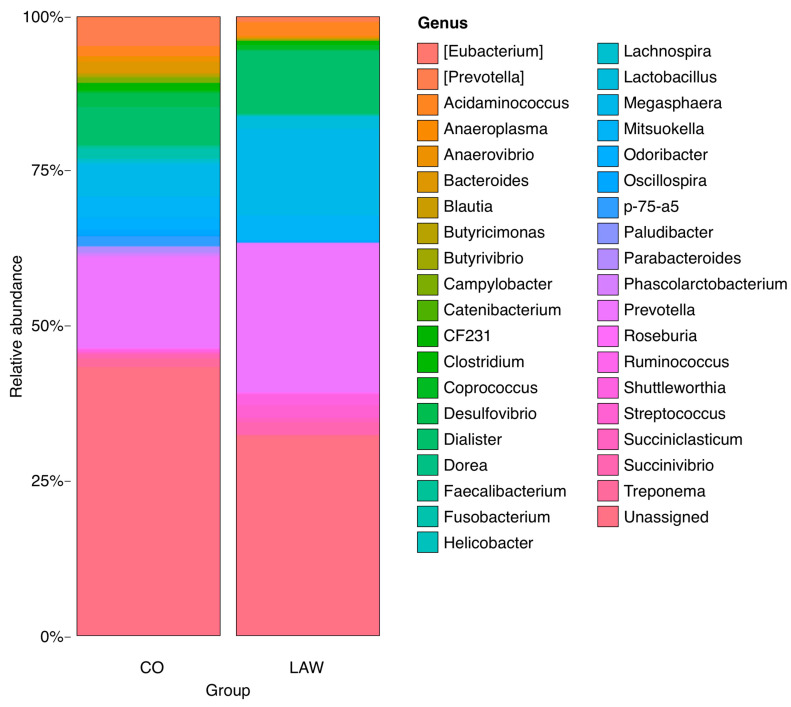
Bar graphs showing proportional abundance of main genus in CO group at left and LAW on the right. A higher relative abundance of *Prevotella* (*p* = 0.00196463024569095), *Anaerovibrio* (*p* = 0.0256783417053815), *Bacteroides* (*p* = 0.00187902637312587), *Butyricimonas* (*p* = 0.00197139594296633), *Campylobacter* (*p* = 0.00342874651022508), *Catenibacterium* (*p* = 0.027559127653885), *Desulfovibrio* (*p* = 0.0256547699171202), *Fusobacterium* (*p* = 0.00631375918655783), *Oscillospira* (*p* = 0.0351950848989481), *p-75-a5* (*p* = 0.00153997525783769), *Parabacteroides* (*p* = 0.00704657239034717) at CO group and *Eubacterium* (*p* = 0.00279865510717577), *Dialister* (*p* = 0.010705311873033), *Lactobacillus* (0.00175250848921009), *Megasphaera* (*p* = 0.00248543353268081), *Odoribacter* (*p* = 0.00142985438326551), *Shuttleworthia* (*p* = 0.0238202172795873), *Streptococcus* (*p* = 0.0149386031285565) in the LAW group.

**Figure 11 animals-13-02611-f011:**
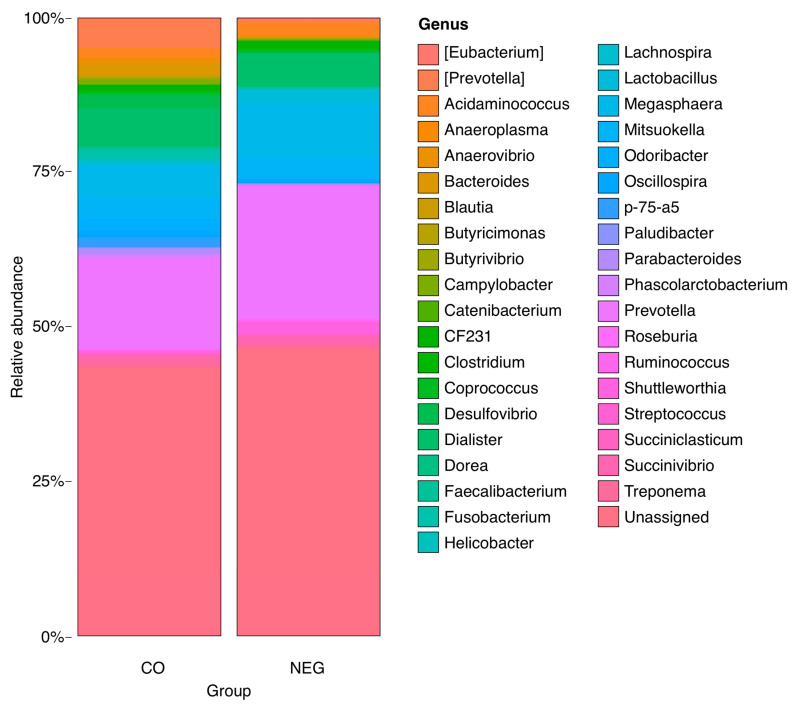
Box plot graphs showing proportional abundance of main genus in CO group on the left and NEG on the right. A higher relative abundance of *Eubacterium* (*p* = 0.037417959614756), *Lactobacillus* (*p* = 0.0127092847550734), *Megasphaera* (*p* = 0.0301176534687101), *Odoribacter* (*p* = 0.000422760407398187), *Shuttleworthia* (*p* = 0.0127092847550734) in NEG group and *Prevotella* (*p* = 7.77198751354089 × 10^−9^), *Anaerovibrio* (*p* = 0.0028415273840731), *Bacteroides* (*p* = 0.00860879727082779), *Butyricimonas* (*p* = 0.01045624760529), *Clostridium* (*p* = 0.0474483843880802), *Desulfovibrio* (*p* = 0.00594276612246729), *Fusobacterium* (*p* = 0.00631375918655783), *p-75-a5* (*p* = 0.00153997525783769) in the CO group.

## Data Availability

The data presented in this study are available on request from the corresponding author.

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
