# Peer review of "Synergic Effect of Brachyspira hyodysenteriae and Lawsonia intracellularis Coinfection: Anatomopathological and Microbiome Evaluation"

_animals, 2023, doi:10.3390/ani13162611_

Round 1
Reviewer 1 Report
1. What is' dpi '? Please provide the full name.
2. The introduction is extremely short and it is best to provide a more comprehensive introduction to the topic.
3. Multiple punctuation and space errors are seen in the manuscript, please carefully check. For example, Line 165 and 312, “..”; “composite score.”; “a,b”.
4. The “*” in figure 1 and figure caption is different.
5. “Arrows indicate mortality of animals”, where are the arrows and how does it indicate mortality of animals.
6. In figure 2, “(d) Animals from LAW and (d) NEG groups with macroscopic normal colon”, there two groups of large intestines in figure (d)? Please clearly mark and separate. The same group is placed in a subgraph, and different groups are separated. Please mark the lesions area with a special symbol.
7. In figure 3 and 4, please provide the original histological and immunohistochemical images as well as scoring method, and indicate the location of the lesion.
8. “The letters “a,b”indicate statistical difference p≤ 0.05”, please clarify how a and b represent significant differences.
9. The figure caption should be more detailed.
10. In figure 6, it is not clear which group the letter b represents in 10/12/15/18dpi.
11. This study aims to compare the differences between co-infection and single infection, and why microbiome of only two groups were analyzed?
12. Please include supplementary figures representing important results of microbiome analysis in the main text.
13. Please improve the clarity of the supplementary figures, as the names of different groups cannot be clearly seen.
14. In figure S5, there are three PCoA Unweighted and Weighted, are they the results of different dpi or others? please mark them clearly.
15. There are too many references, please cite them appropriately.
Author Response
Answer to reviewer 1 are attached to the system

Reviewer 2 Report
The manuscript by Daniel et al. describes synergisms in experimentally infected pigs of four groups of piglets: coinfection (CO), B. hyodysenteriae (BRA), L. intracellularis (LAW), and negative control (NEG). The paper is well written. The data are very informative since clinical data, gross lesions, bacterial isolation, histopathology, immunohistochemistry, and fecal microbiome analyses were performed. Altogether, data conclude that animals co-infected with B. hyodysenteriae and L. intracellularis showed more severe clinical signs as well as gross and histological lesions than animals infected with only one of these agents. I think this is a very well-organized paper that would be very attractive to the readers of this journal.
Minor comments:
Line 24: please correct.
Lines 107 and 121: could the authors explain why the infectious dose was 10^6? And Line 365: “which is considered low compared to studies in which there was a typical clinical manifestation (108 – 1010) [31, 45].” Is it known the infectious dose in real field conditions?. Some discussion about this would benefit the paper
Line 142: qPCR analysis included a standard curve. Could please the authors explain how was this standard curve performed?
Line 265 “In the NEG group, two animals had a diarrhea score of 3” could the authors give some explanation to understand why animals in this group showed diarrhea (in the NEG group, also one animal presented moderate hyperemia of the colonic serosa)?
Line 311: L. intracellularis. Italics
Line 404: “Previous studies evaluated the microbiome of animals inoculated with B. hyodysenteriae or L. intracellularis have been previously described in some studies [20, 62-63].” Check grammar
Line 440: “The increase in Desulfovibrio was also shown Burrough et al. [63] who observed an increase in Desulfovibrio in swine with dysentery.”. Check grammar
-
Author Response
Answers to reviewer 2 are attached in the system

Round 2
Reviewer 1 Report
No